# Are We Studying Non-Coding RNAs Correctly? Lessons from nc886

**DOI:** 10.3390/ijms23084251

**Published:** 2022-04-12

**Authors:** Yong Sun Lee

**Affiliations:** Department of Cancer Biomedical Science, Graduate School of Cancer Science and Policy, National Cancer Center, Goyang 10408, Korea; yslee@ncc.re.kr; Tel.: +82-31-920-2748; Fax: +82-31-920-2759

**Keywords:** non-coding RNA, nc886, overexpression, knockdown, assay

## Abstract

Non-coding RNAs (ncRNAs), such as microRNAs or long ncRNAs, have brought about a new paradigm in the regulation of gene expression. Sequencing technologies have detected transcripts with tremendous sensitivity and throughput and revealed that the majority of them lack protein-coding potential. Myriad articles have investigated numerous ncRNAs and many of them claim that ncRNAs play gene-regulatory roles. However, it is questionable whether all these articles draw conclusions through cautious gain- and loss-of function experiments whose design was reasonably based on an ncRNA’s correct identity and features. In this review, these issues are discussed with a regulatory ncRNA, nc886, as an example case to represent cautions and guidelines when studying ncRNAs.

## 1. Introduction

Non-coding RNAs (ncRNAs) are RNA molecules without protein-coding potential. The central dogma of modern molecular biology indicates that the flow of genetic information is from DNA, via RNA as an intermediate, to proteins that ultimately have catalytic roles and thus, are responsible for most cellular metabolism. Therefore, in the authentic central dogma viewpoint, ncRNAs could even be considered to be odd. Currently, the ubiquitous existence of ncRNAs has become common knowledge. The majority of the human genome is transcribed, and most transcripts are non-coding [1]. Furthermore, during the last two decades, ncRNAs have emerged as important regulators for gene expression. This discovery caused a big shift in our concept about ncRNAs. The conventional concept was that ncRNAs are constitutively expressed and play fundamental roles, when we recall classic ncRNAs such as transfer RNAs (tRNAs), ribosomal RNAs (rRNAs), small nuclear RNAs (snRNAs), etc. Although some pioneering researchers had earlier conceived the idea of ncRNA–mRNA interaction and its impact on mRNA expression [2,3,4], the research trend in the study of “regulatory ncRNAs” began in earnest in 2001, when microRNAs (miRNAs) were found to be ubiquitously present [5,6,7]. This trend was accelerated and widened by the development of next-generation sequencing (NGS) technologies [8,9]. By virtue of its unprecedented sensitivity and throughput, NGS captured a myriad of ncRNAs that had been undetected before. There has been a debate about whether they are functional molecules or merely transcriptional noise. In numerous articles, researchers claimed that they found a biological role for the ncRNAs that they investigated. Nonetheless, it is questionable whether all research articles convey genuine information through correctly performed experiments and proper interpretation of data. 

Therefore, in this critical review, I would like to discuss the caveats of ncRNA studies, based on my experience while investigating ncRNAs, especially nc886, which has been my main research theme for over ten years. Based on lessons from nc886, I want to suggest several points to be cautious of in designing experiments and interpreting data. Although a significance number of references have been critically cited here, I should mention that they convey valuable information and have contributed to the growth of our knowledge. Importantly, most of them have precisely described experimental procedures so that I could rely on the data and interpret them correctly. I welcome any comments from the authors and readers and am open to constructive discussion.

## 2. Lessons from nc886

### 2.1. nc886 Had Been Misidentified

nc886 is a 101 nucleotide (nt) long ncRNA whose aliases include vault RNA2-1, pre-miR-886, etc. As one of its aliases indicates, nc886 was mis-annotated as an miRNA precursor, and its mature miRNAs, miR-886-5p and -3p, had been registered in the miRNA database (miRBase releases 10–16, https://www.mirbase.org/, accessed on 1 March 2022; Figure 1). In fact, my initial interest in nc886 came from miRNA array data, which yielded miR-886-5p and -3p to be most differentially expressed between malignant and non-malignant cells [10]. However, experimental evidence from my laboratory led to a conclusion that nc886 is not an miRNA [10]. The issue of whether they are functional miRNAs is very important, because most papers on miR-886-5p and -3p employed miRNA study tools (to be elaborated later). If they are not miRNAs, those data should be reexamined.

miRNAs are defined to be ~21 nt-sized small RNAs that are produced from a hairpin precursor (pre-miRNA) by Dicer [11]. A long primary transcript is processed by Drosha to generate a pre-miRNA, which is cleaved by Dicer and is loaded to Argonaute (Ago) family proteins to suppress the expression of target mRNAs (reviewed in [12]). When recognizing target mRNAs, the critical sequence element of an miRNA is a so-called “seed sequence”, which is 6–8 nt at the 5′-side of an miRNA (position 1~2 to 7~8) (reviewed in [13]). Small RNA sequencing (small RNA-seq) by NGS captured RNA fragments corresponding to miR-886-5p and -3p sequences, providing evidence that they exist [14,15,16,17]. However, the existence of RNA fragments does not guarantee that they are real miRNAs, because there are various types of small RNAs and miRNA is one of them (reviewed in [18]). Currently available data, such as Northern blot and miRNA activity assays, unequivocally indicate that nc886 does not produce miR-886-5p or -3p at a level that could be significant as a functional miRNA [10,19,20,21,22,23,24]. Northern blot in all these papers failed to detect a ~21-nt band. Even when detected, the bands were extremely minute in quantity and appeared to be smeared [25,26], indicating that they are degradation products of nc886. In fact, the steady-state level of nc886 is very abundant, despite its short half-life [10,27], leading to an expectation that a significant amount of degradation products are continuously produced (Figure 1).

In conclusion, RNA fragments corresponding to miR-886-5p or -3p exist, but they are most likely to be degradation products whose level is negligible compared to the intact, 101 nt long nc886 (Figure 1). Nonetheless, there are a number of papers about miR-886-5p or -3p (40 articles retrieved in the PubMed database when searched by “miR-886-3p OR miR-886-5p”). I will discuss here whether data in the literature could draw a conclusion that miR-886-5p or -3p are functional miRNAs.

### 2.2. What Is Measured versus What Really Exists; Are They the Same?

The majority of the papers on miR-886-5p or -3p came from an attempt to find an miRNA that played a role in a biological situation of their interest. miRNA study typically begins with array experiments to screen an miRNA whose expression levels are altered. In hybridization-based array platforms, probes for miR-886-5p or -3p will also detect nc886. PCR-based arrays, which usually employ the TaqMan PCR technique [28], cannot distinguish nc886 from miR-886-3p in principle, since they share an almost identical 3′-end (Figure 1). In agreement with this notion, a significant fraction of studies employing TaqMan array platforms claimed to preferentially detect miR-886-3p rather than -5p [29,30,31,32,33]. What they detected was probably nc886, as inferred from the aforementioned Northern blot results [10,19,20,21,22,23,24,25,26]. Hence, a positive signal in array platforms cannot prove the existence of miR-886-5p or -3p. This would be easily understandable if compared with conventional array experiments to measure gene expression. Array probes are an oligonucleotides complementary to the target mRNA. From the positive hybridization signal, everyone notices the mRNA expression level, but nobody would insist that an RNA fragment corresponding to the probe sequence exists naturally. In many cases of hybridization-based arrays, miR-886-5p and -3p exhibited a similar expression pattern [34,35,36,37]. These data strongly indicate that nc886 was what they detected, similar to the case of conventional arrays, where an mRNA is detected by multiple probes.

### 2.3. The Mistaken Identity Misleads Functional Approaches

When researchers have selected an miRNA of their interest, the next step is typically to examine cellular phenotypes and target mRNAs by performing overexpression or knockdown (KD) experiments. The most prevalent method is to transfect an miRNA mimic or inhibitor. Caution is required, especially when using an miRNA mimic. It should be noted that miRNA mimics are a chemically synthesized RNA duplex, whereas natural miRNAs are single-stranded (Figure 1).

To understand this discrepancy, a story about miRNA and RNA interference (RNAi) needs to be told (reviewed in [38]). In the RNAi pathway, long double-stranded RNA (dsRNA) is cleaved by Dicer to produce small interfering RNAs (siRNAs) that are a ~21 nt duplex with a 2 nt overhang at the 3′-ends. A duplex with such an end structure is exactly the Dicer/Drosha processing product in the miRNA pathway [12]. The miRNA and RNAi pathways merge at the Dicer step to produce miRNA and siRNA duplexes with nearly identical structure, and then both of them are loaded to Ago proteins to suppress target mRNAs. When loaded to Ago proteins, only one strand (called the “guide strand”) survives to recognize target mRNAs, but the other strand (the “passenger strand) is degraded. Once loaded on Ago proteins, it is not distinguishable whether the RNA was originally from an miRNA or an siRNA. This is the reason for an siRNA’s off-target effect, as first documented by the Dutta laboratory [39]. Although an siRNA is designed to be perfectly complementary to its target mRNA, it could suppress a number of unintended mRNAs (off-targets) with a complementary sequence to the seed region (positions 1~2 to 7~8 of an miRNA) via an miRNA mechanism [38]. Since the seed region is only 6–8 nt long, the probability of appearance is once per ~4 to 64 kb, which estimates more than several tens of thousands of target sites in the human genome sequence. This estimation, albeit a simple arithmetic calculation, indicates that any siRNA is likely to have a significant number of off-targets.

The above story provided a theoretical background to justify the use of an siRNA-like duplex as the functional mimic of a miRNA [40]. Although very briefly described above, there were a significant number of research endeavors to establish this miRNA overexpression method. In addition, the story gives an important note of strict warning that an siRNA-like duplex must be used only for bona fide miRNAs. If an siRNA-like duplex is used for overexpression of a small RNA which is not an miRNA, it will lead to suppression of off-target mRNAs, which is irrelevant to the small RNA’s genuine biological role. As a result, the experimental data will be entirely artifactual. miR-886-5p and -3p are such examples (Figure 1).

From overexpression data, several papers claimed that they found a cellular role of miR-886-5p or -3p and identified target mRNAs [16,17,25,41,42,43,44,45,46]. However, any experimental data obtained from the transfection of miR-886-5p or -3p mimics do not reflect their natural role nor prove their existence. For easy understanding, I want to present a suppositional situation that a researcher uses an siRNA against a gene, for example, TP53 (encoding p53, a tumor suppressor protein). Transfecting the siRNA will lead to KD of TP53 and other off-target genes to elicit a resultant phenotype. However, this does not prove their biological existence nor a natural role of “a small RNA antisense to TP53”. Nobody would be interested in it, either. Likewise, an miR-886-5p or 3p mimic is an siRNA-like synthetic duplex; thus, it will suppress some off-target genes to cause a certain phenotype, when transfected into cells (Figure 1). However, this cannot be evidence for the natural existence or role of miR-886-5p or -3p, as in the case of “a small RNA antisense to TP53”. Some studies imply that a phenotype resulting from the transfection of an miR-886-5p or -3p mimic represents the gain-of-function of nc886, because they claim that miR-886-5p or -3p is derived from nc886 [16,17]. However, an miR-886-5p or -3p mimic might lead to a loss-of-function by acting as an siRNA targeting nc886 (Figure 1). In fact, one report used an siRNA for nc886 KD [47].

In summary, from data that have been published so far, nc886 does not seem to produce functional miRNAs; experimental evidence is not sufficient to support an miRNA role for miR-886-5p or -3p. Overexpression data from a synthetic duplex should be interpreted with great caution (Figure 1). I do not rule out a possibility that nc886 might produce functional miRNAs in certain biological situation. Studies with Ago-association data, one on Parkinson disease and the other on prostate cancer, might have found such situations [16,17]. However, to prove a role of miR-886-5p or -3p fragments as an miRNA, more experimental data are required, but are yet to be available. Such data include overexpression by a plasmid-based method (see the next section), a KD experiment, etc.

### 2.4. Features of ncRNA also Matter When Designing Gain-of-Function Experiments

Plasmid vectors are the most common tool for the ectopic expression of a gene. nc886 is silenced in a number of cancer cell lines. When attempting its ectopic expression in these cell lines, the vector-based method should certainly be the primary choice. In this strategy, knowledge of nc886 is needed for optimal experimental design.

nc886 is transcribed by RNA polymerase III (Pol III), but not by RNA polymerase II (Pol II) [26,27,48,49,50]. Genes transcribed by Pol III (shortly, “Pol III genes” or “Pol III transcripts”) are classified into three types, according to cis-acting promoter elements that determine which initiation subunits of the Pol III enzyme to be recruited (reviewed in [51]). Representative genes for type 1, 2, and 3 are 5S rRNA, tRNAs, and U6 snRNA, respectively. nc886 contains gene-internal promoter elements, A and B boxes, which resemble those of type 2 (Figure 1). Some type 2 Pol III genes, such as vault RNAs (vtRNAs), 7SL RNA, and BC200, also require their 5′-upstream sequence, in addition to A and B boxes, and these genes are sub-classified as type 2H (reviewed in [52]). nc886 is a paralog of vtRNAs and is supposed to belong to the type 2H. Actually, in my unpublished data, nc886 was efficiently expressed from a plasmid devoid of a mammalian promoter only when > 200 nt at the 5′-upstream was inserted together with the nc886 RNA region. This was one way to construct an nc886-expressing plasmid (Figure 1). Another way was to insert the nc886 RNA region under the U6 or H1 promoter, which is a type 3 Pol III promoter ([53,54], Figure 1). Plasmids constructed according to these two strategies expressed nc886 correctly, as indicated by a single band at the identical size of the endogenous nc886 in Northern blot [10,22,27,54]. Moreover, when stable cell lines expressing nc886 were made with these plasmids, the ectopic expression level of nc886 was comparable to the endogenous levels of cell lines naturally expressing nc886 [22]. These data ascertained the legitimacy of the ectopic expression and the reliability of the resultant phenotypes.

Commonly used expression vectors have a strong promoter, which is usually adopted from viruses such as cytomegalovirus (CMV) or simian virus 40 (SV40). A couple of studies employed these vectors to construct nc886-expression plasmids [45,55]. It should be pointed out that the CMV or SV40 promoter is for Pol II (Figure 1). Most likely, Pol II will ignore a TTTT sequence, the termination signal for Pol III transcription, at the 3′-end of nc886. It is also questionable whether the transcription driven by an entirely irrelevant promoter would start at the correct position. Consequently, those plasmids are presumed to yield longer transcripts harboring the nc886 sequence (Figure 1). These extended transcripts cannot be distinguished from the correct nc886 of 101 nt, if qRT-PCR is employed as a method to confirm overexpression. I do not rule out a possibility that those Pol II transcripts might be capable of mimicking nc886 functionally, because they contain the nc886 sequence. Nevertheless, certainly there exists a risk that it cannot present nc886′s function, for a number of plausible reasons. These long, Pol II transcripts are likely to undergo a path different from the Pol III-transcribed nc886. As a result, they might lack necessary post-transcriptional processing steps or be mis-localized within a cell. Appending sequences in the longer nc886 transcripts might interfere with the formation of the nc886′s correct secondary structure.

In one report, a sufficient length (>200 nt) of the 5′-upstream sequence, together with the nc886 RNA region (101 nt), was inserted downstream of a Pol II promoter [45]. Such a plasmid is expected to drive transcription of the correct nc886 RNA, in addition to long Pol II transcripts. Nonetheless, there are still concerns. A potent Pol II promoter might keep recruiting the Pol II enzyme complex, which competes with the Pol III enzyme complex for occupation on DNA, resulting in low-level expression of the correct nc886. Longer Pol II transcripts might inhibit the correct nc886 in a dominant-negative manner (Figure 1). These concerns indicate a need to conduct Northern blot or any functional assay to confirm whether the correct nc886 is expressed at a sufficient quantity.

### 2.5. Choosing a KD Method

RNAi-based methods and modified antisense oligonucleotides (ASO) are commonly used for KD of a gene (reviewed in [56]). RNAi-based KD is usually highly effective for protein-coding genes, but ineffective for some ncRNAs [57]. In addition, for nc886, RNAi-based method does not seem to be effective according to a report [47] in which they transfected an siRNA against nc886 and measured the activity of protein kinase R (PKR), a protein that nc886 normally suppresses [10,58,59,60]. PKR is typically activated by long dsRNAs (>55 nt) that are generated during viral infection. Since PKR should be autophosphorylated for its kinase activity, Western blot analysis of a phosphorylated form of PKR (phospho-PKR) is indicative of its activation [61]. The transfection of siRNA against nc886 led to an increase in phospho-PKR, indicating nc886 KD [47], but the fold-increase was very modest as compared to those in dsRNA or legitimate KD of nc886 (see below).

The standard method for nc886 KD in my laboratory is to transfect a 20 nt long ASO, having five 2′-O-methyl ribonucleotides at both ends [10]. The backbone is phosphorothioate-modified. This design is a DNA-RNA mixmer, based on RNaseH-mediated cleavage of the heteroduplex between a target RNA and the middle DNA portion of the ASO [62,63]. The transfection of this ASO into various cell lines led to efficient KD, as indicated by the measurement of the nc886 and phospho-PKR. In most cases, the 101 nt nc886 band was clearly decreased in Northern hybridization, and phospho-PKR was robustly increased to a comparable degree by its canonical activator dsRNA [10,19,20,22,64,65,66,67].

Because it was mis-identified as an miRNA, several papers intended the KD of miR-886-5p or -3p using commercially available miRNA inhibitors [25,41,44,68]. These inhibitors are also modified ASOs and thus, could have suppressed nc886. However, in these papers, nc886 was not their purpose and thus, was not measured. These miRNA inhibitor ASOs are worth considering for nc886 KD, although the DNA-RNA mixmer works well in most cell lines.

### 2.6. Are We Evaluating KD Efficiency Correctly? Importance of Assays

When choosing a KD method in actual experiments, the measurement of KD efficiency by reliable assays was more important than the theoretical principle of a KD method. I was surprised to see that the DNA-RNA mixmer ASO was highly efficient for nc886 KD. Conceivably, RNaseH-mediated cleavage will occur dominantly in the nucleus to destabilize a target RNA [69]. On the contrary, nc886 is localized exclusively in the cytoplasm and is intrinsically unstable, with a half-life of ~one hour [10,22,27]. I did observe almost complete disappearance of nc886, which was hardly imagined if the mechanism was the destabilization of “already unstable” RNA. The actual mechanism of how an ASO works in transfected cells appears to be more complicated than a theoretical method based on the ASO design chemistry, and it may even differ among cell lines. In most cell lines, nc886 expression was decreased by the DNA-RNA mixmer ASO. However, there was an exceptional case in the cholangiocarcinoma cell lines, in which the same ASO did not decrease nc886, but induced PKR activation [64]. Further investigation, by immunoprecipitation experiments, elucidated that the ASO led to the dissociation of nc886 from PKR, leading to PKR activation without affecting the nc886 RNA level.

These nc886 episodes highlight the importance of assays. Without knowing nc886′s role in PKR, anyone would have concluded that KD failed in the cholangiocarcinoma cells. It should be also noted that the examination of nc886 by Northern hybridization was the most convincing way to validate the KD efficiency. In qRT-PCR assays, I would have a concern about a false-positive result where inefficient KD is mistakenly assessed to be successful. A huge amount of an ASO could interfere with cDNA synthesis or PCR amplification, since the ASO will exist in the RNA preparation and possibly re-associate with the target RNA during these enzymatic steps. To avoid this possibility, the best way will be to design an ASO outside of a PCR amplicon. However, such a design may yield a false-negative result, in which KD is actually successful, but looks like a failure. Another complication is that an ASO could inhibit the activity of an ncRNA without affecting its expression level, as in the case of nc886 in the cholangiocarcinoma cells [64]. Therefore, an ideal way to assess KD efficiency would be to use a functional assay in combination with a proper method for RNA measurement, such as Northern hybridization.

## 3. Suggestions from the nc886 Lesson, When Studying an ncRNA

Based on the experiences of nc886, I would like to bring some points to attention when studying an ncRNA.

### 3.1. Are We Looking at ncRNA Molecules Correctly?

As compared to protein-coding RNAs, extra precautions are required when defining an ncRNA’s identity because the RNA molecule itself is a functional entity.

The primary element of identity is a precise RNA sequence from the 5′-end to the 3′-end. In the case of protein-coding genes, the functional entity is the translation products, and their sequences are usually unambiguously defined by start and stop codons. This information on DNA can be complemented by peptide sequencing. In the case of ncRNAs, it is highly questionable whether their sequences in public databases represent their precise 5′- and 3′-ends. I was surprised to see that even the nc886 sequence is incorrectly displayed in one of the most authoritative databases, the National Center for Biotechnology Information (NCBI) database (Figure 1). nc886’s correct 3′-end is undoubtedly “…TTTT”, since it is the Pol III termination signal, and it was experimentally validated by the rapid amplification of the cDNA end (RACE)-PCR technique [10]. The NCBI reference sequence (NR_030583.3) shows six extra nts at the 3′. This incorrect information might mislead experimental designs.

The molecular shape of an ncRNA is determined by other features beyond its precise nt sequence. These include a 5′-end structure (mono- or tri-phosphorylated, canonically capped, or some other type), an nt addition at the 3′-end (oligo- or poly-adenylation or uridylation), and base modification, as in the case of tRNAs. All these factors can determine an ncRNA’s function. When most research laboratories study an ncRNA, initial screening comes from NGS data, and validation is done by PCR-based techniques. Strictly speaking, NGS and PCR methods measure only a part in an entire ncRNA molecule, and not any of features described above.

### 3.2. Are We Properly Performing Gain- and Loss-of-Function Approaches Based on Correct Identity and Features?

To overexpress an ncRNA, a common method is to clone an ncRNA sequence referred from databases into an expression vector. As mentioned earlier, sequences in databases cannot guarantee their genuine 5′- to 3′-end, presumably in many cases. I would like to state again that, for ncRNAs, it is an RNA molecule itself that functions. For better understanding, we can compare RNA with a protein. Do you expect a chance that a truncated polypeptide would phenocopy the full-length wild type protein? Rather, one has to be concerned about the possibility that the truncated polypeptide might suppress the wild type protein in a dominant-negative manner.

Not only the identity, but the quantity also matters. The natural expression levels of most ncRNAs are very low [70]. When this expression is driven by a mighty promoter, such as SV40 or CMV, an ncRNA will be expressed at a supra-physiological level, raising a concern that the phenotypic outcome might be an overexpression artifact.

In the case of small ncRNAs which are not miRNAs, the transfection of a synthetic RNA can be considered for overexpression. The easiest design would be a plain single-stranded RNA oligonucleotide, but they are usually fragile when introduced into cells [71]. Therefore, backbone modification might be required. In this case, RNA measurement is meaningless, because the synthetic RNA is provided into the cells and will be detected anyway, regardless of whether they are functional or not. A functional assay is absolutely required.

Functional assays in the cancer cell biology field, as an example, include cell proliferation, apoptosis, colony formation, migration, and invasion assays. Although these are routinely performed assays, the measurement of specific genes, protein activities, or pathways (for example, induction of phospho-PKR by nc886 KD) is more desirable as a functional assay. Gene expression profiling data upon overexpression and KD/KO of an ncRNA will provide a hint as to which should be measured, and further, a clue about its molecular function.

For loss-of-function approaches, I would consider modified ASO as a higher priority than siRNA. The adverse effects of an siRNA (off-target activity and saturation of the miRNA machinery) are obvious [38,72]. There are numerous types of ASO [56] and the most effective one should be determined empirically, based on proper assays. For the reasons extensively discussed in the case of nc886, Northern hybridization is superior to qRT-PCR. Besides RNA measurement, it is desirable to perform a functional assay. However, a functional assay is unavailable in many cases, because the purpose of KD of an ncRNA is to find its function.

A gene editing technique called clustered regularly interspaced short palindromic repeats (CRISPR)-Cas (CRISPR-associated proteins) has provided a method for gene knockout (KO) [73]. KO experiments have an intrinsic shortcoming, since KO cells cannot be made for a gene’s essential cell growth. Nonetheless, once KO cells are successfully created, they will provide the most convincing data for an ncRNA’s role.

## 4. Summary of Cautions for Studying an ncRNA

### 4.1. A Prerequisite for Studying an ncRNA Is to Know Its Features, Including

-What is its exact sequence? The precise 5′- to 3′-end should be determined in order to correctly design overexpression and KD experiments.-Which RNA polymerase (I, II, or III) transcribes it?-Where is it located (nuclear or cytoplasmic, in a protein complex or chromatin-associated, in a specific subcellular organelle, etc.)?-Is it a de novo transcript or a processing product from a pre-existing RNA?-How abundant is it in a cell?

### 4.2. Gain-of-Function Experiments

-In a plasmid-based overexpression system, the precise sequence should be placed under a correct promoter of reasonable strength.-For small ncRNAs, a synthetic modified oligoribonucleotide can be considered.-For small ncRNAs without concrete evidence that they are bona fide miRNAs, an siRNA-like duplex must not be used.

### 4.3. Loss-of-Function Experiments

-For KD, modified ASO should be considered with a higher priority than an siRNA.-KO by CRISPR-Cas would be the best, if KO cells can be created.

### 4.4. Assays for Overexpression or KD and Assessment of Cellular Phenotypes

-Overexpression or KD should be validated by a proper measurement; Northern hybridization is better than qRT-PCR.-In addition to measurement, validation by a functional assay is desirable.-A functional assay is absolutely required when a synthetic RNA is used for overexpression.-A functional assay can include the use of cellular phenotypes, gene expression profiles, or the measurement of a particular protein activity or pathway. The more specific, the better.-The phenotypic data from overexpression and from a KD or KO experiment should complement each other.

## 5. Concluding Remark

ncRNAs have been a prevailing research topic. Numerous ncRNAs have been detected and accordingly, the ncRNA field has grown enormously, with approximately 200,000 papers published so far. Among these, nearly three-quarters are miRNA papers, owing to handy tools for miRNA research: miRNA detection, overexpression, inhibition, and target prediction are all straightforward. However, for most other types of ncRNAs, it is a challenging task and a long journey to find an ncRNA’s genuine biological role. In case of orphan ncRNAs, it is understandably a difficult job. For an ncRNA in a particular class, it is no easier. I spent a significant amount of effort proving that nc886 (a.k.a. vtRNA2-1) is not an miRNA precursor, nor does it have a vault-related function, although it is known as a vtRNA paralog.

Although those approximately 200,000 published papers have vastly expanded our knowledge, we should be careful in accepting the alleged role of an ncRNA therein. When a researcher obtains a different, or even conflicting, result during the investigation of an ncRNA, it would be worth revisiting the original article with a thorough inspection of the experimental procedures and the critical interpretation of the data, instead of accepting the conclusions of the paper. For on-going studies, evidence should be provided by prudently designed experiments to concretely prove an ncRNA’s role. This will prevent the situation where “bad money drives out good” from occurring in the ncRNA field. I hope that this review contributes to that prevention.

## Figures and Tables

**Figure 1 ijms-23-04251-f001:**
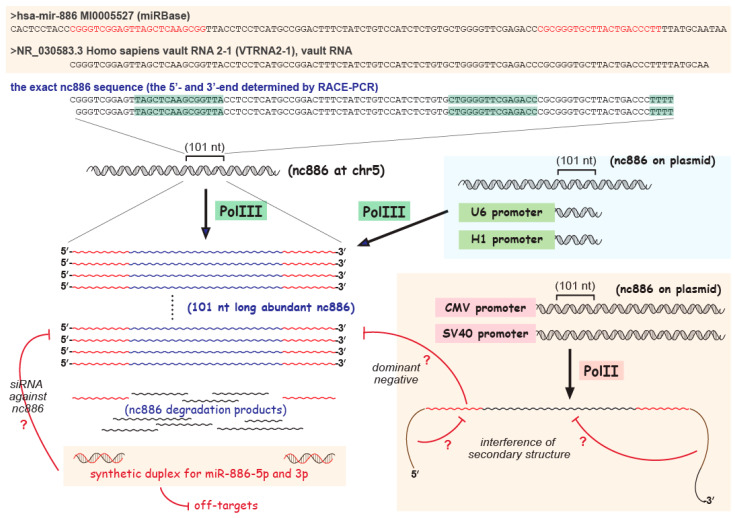
The sequence of nc886 and its natural expression, in comparison to ectopic expression methods. The three sequences on the top are from miRBase (red letters: miR-886-5p and -3p), from the NCBI database, and experimental determination. Pol III promoter and termination elements are highlighted in green shadow. The nc886 RNA is drawn in wavy lines in which red portions designate miR-886-5p and -3p. Improper ectopic expression of nc886 (and its alleged products miR-886-5p and -3p) is illustrated in faint orange boxes. Magenta blunt arrows with a question mark indicate and potential side effects of illegitimate expression. See the main text for details.

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
