# Peer review of "Are We Studying Non-Coding RNAs Correctly? Lessons from nc886"

_ijms, 2022, doi:10.3390/ijms23084251_

Round 1

Reviewer 1 Report

Comments to the Author

The manuscript by Lee, systematically demonstrated that ncRNAs as prevailing research have been greatly developed and made great research progress. Meanwhile, through a large number of researches on nc886, the author cautiously put forward the hope that the researchers can critically view the results of original article, as well as be more rigorous for the current study, to jointly promote the great development of the field. 

Concerns:

  1. The theme of the article is about ncRNAs, but there is a lack of introduction to ncRNA background knowledge.
  2. There are some less relevant discussions in the article, such as lines 132-145. The author can also try to give examples of other ncRNAs to make the content more complete.
  3. Line 207, the author describes KD method in detail here, which is repeated with the previous text and can be simplified appropriately.
  4. It is recommended that the author summarize the article core at the end of the paper and simply list the correct research means of ncRNAs and how siRNAs and miRNAs can be distinguished. This can help readers obtain information faster and thus benefit from this research.
  5. issues with reference format, please modify.
  6. Line 624, there is a issue with the figure legend of Figure1.

Reviewer 2 Report

Specific comments to the authors

The titled review “Non-coding RNAs are fascinating players in gene regulation, but are we studying them correctly?” discuss problematic issues of the non-coding RNAs (ncRNA) and especially “pars pro toto” of the proposed example ncRNA 886 (nc886) in gene regulation.

The presented topics range from specific lessons of nc886 regarding misidentification, measurement and experimental assessments of functionality to suggestions for solving this issues.

In summary, the mentioned review gives interesting insights of some problematic issues of investigation the role of ncRNA in human biology and pathology, which is mostly easy to read, to follow and to understand, overall. Nevertheless, the presented manuscript have more the characteristics of a personal opinion/note (like an editorial comment) than a real review. Furthermore, the author should clarify some aspects before accepting the manuscript for publication as mentioned below.

# Title: As the author focuses on ncRNA 886, the title should contain a reference to it.

# Synopsis: This section is largely descriptive and does not really lead to the definitive issues and aspects of the Non-coding RNAs.

# “1. Lessons from nc886”: The statement “What they detected was probably nc886” should be proven by more evidences. The statement of “any siRNA is likely to have a significant number of those off-target mRNAs” is largely unspecific. Regarding the sentence “Transfecting the siRNA will lead to KD of p53 and other off-target genes to elicit a resultant phenotype.“ and following it should kept in mind that researchers do normally choose more siRNA variants to enhance the specifity/sensitivity, overall. The pro and contras of gain-of-function-experiments (using vector-based method) should be gathered in a separate table for better reading and understanding.

# “2. Suggestions from the nc886 lesson, when studying an ncRNA”: This chapter repeats most of the previous chapter merged with hints for “correct” study of non-coding RNAs. Therefore, please define clearly the suggestions to follow in experimental setting (perhaps with an additional figure).
